# Development, Optimization and Validation of a Sustainable and Quantifiable Methodology for the Determination of 2,4,6-Trichloroanisole, 2,3,4,6-Tetrachloroanisole, 2,4,6-Tribromoanisole, Pentachloroanisole, 2-Methylisoborneole and Geosmin in Air

**Patricia Jové** [1],*, **Marina Vives-Mestres** [2], **Raquel De Nadal** [3] **and Maria Verdum** [1]

1   Department of Research and Development, Catalan Cork Institute and Foundation (Institut Català del Suro, ICSuro), 17200 Girona, Spain; mverdum@icsuro.com
2   Department of Computer Science, Applied Mathematics and Statistics, Universitat de Girona, 17200 Girona, Spain; marina.vives@udg.edu
3   Department of Research, J.VIGAS S.A, Palafrugell, 17200 Girona, Spain; raquel@jvigas.com
*   Correspondence: pjove@icsuro.com; Tel.: +34-972-305-661

**Abstract:** Compounds 2,4,6-trichloroanisole (TCA), 2,3,4,6-tetrachloroanisole (TeCA), 2,4,6-tribromoanisole (TBA) and pentachloroanisole (PCA), 2-methylisoborneol (2MIB) and geosmin (GSM) have been reported as being responsible for cork and wine taint. A sustainable method based on thermal desorption-gas chromatography–mass spectrometry (TD-GC/MS) has been developed and optimized, taking into account desorption parameters and chromatographic and mass spectrometric conditions. The combination of parameters that jointly maximized the compound detection was as follows: desorption temperature at 300 °C, desorption time at 30 min, cryo-temperature at 20 °C and trap high temperature at 305 °C. The proposed methodology showed a good linearity (R ≤ 0.994) within the tested range (from 0.1 to 2 ng) for all target compounds. The precision expressed as repeatability and reproducibility was RSD < 10% in both. The limits of quantification ranged from 0.05 to 0.1 ng. The developed methodology and the sampling rates (R-values) of all targeted compounds (from 0.013 to 0.071 $m^3\ h^{-1}$) were applied to the air analysis of two wineries. The results showed that the developed methodology is a sustainable and useful tool for the determination of these compounds in air.

**Keywords:** off odours; haloanisoles; wine taint; winery; TD-GCMS; air monitoring

## 1. Introduction

Haloanisoles (HAs): 2,4,6-trichloroanisole (TCA), 2,3,4,6-tetrachloroanisole (TeCA), 2,4,6-tribromoanisole (TBA) and pentachloroanisole (PCA) are well-known wine contaminant compounds that cause unpleasant musty aromas. HAs are very volatile, easily transmitted through the air and highly capable of contaminating wine, wood, cork and many other materials. For years, the wine industry thought that HAs were exclusively associated with cork stoppers, but many studies have reported other sources of HAs contamination such as the use of pesticides and wood preservatives containing halophenols (HPs). Geosmin (GSM) and 2-methylisoborneol (2MIB) are produced as secondary metabolites by specific microorganisms, including actinomycetes, cyanobacteria (blue–green algae), myxobacteria and fungi. All of these are considered responsible for earthy–musty odours in drinking water and wine [1–5] and represent a serious economic problem for wineries.

In general, early detection of these compounds in wine, in auxiliary materials involved in the winemaking process or in air, is very important to prevent the contamination of wine [6]. A number of methods have been proposed for the determination of HAs, GSM and 2MIB but they have focused on cork stoppers, water or wine [5,7–9]. Due to the physical and chemical properties of these compounds, gas chromatography coupled to mass



spectrometry (GC–MS or GC-MS/MS) has been the preferred technique for their detection and quantification. For solid and liquid samples (cork stoppers or wine), solid phase microextraction (SPME) and solid phase extraction (SPE) have been the most commonly used extraction techniques [8,10,11]. The only procedure to qualitatively determine HA in air is by using bentonite as a passive adsorbent. In this case, after a few days of bentonite exposure in air, a certain quantity is extracted with hexane and later analyzed using GC techniques. The results are expressed as ng of the compound per g of bentonite. Therefore, it is a qualitative method.

In order to promote a more sustainable and quantifiable methodology, an effective method based on thermal desorption coupled to gas chromatography (TD-GCMS) is proposed for the determination of low concentrations of selected compounds in air. In this case, targeted compounds are adsorbed onto porous polymers for subsequent analysis using TD-GCMS [12–16]. TD-GCMS allows the detection and quantification of the selected compounds in air using active or/and passive sampling procedures. Although both procedures provide good performance and great flexibility, passive sampling has gained popularity due this procedure not needing a calibrated pump to collect samples. In this case, compounds from the air accumulate onto the sampling medium via diffusion. For this reason, it is essential to determine the sampling rates or diffusive uptake rates (R-value) for each targeted compound. This value is a characteristic of each compound related to its physicochemical properties and can also be influenced by environmental conditions such as air temperature, humidity or sampler design [17,18]. Some studies have determined R-values in the laboratory using artificially polluted chambers or reactors [17,19–23]. Camino et al., 2015 [13] determined the R-values of TCA and trichlorophenol (TCP) for monitoring air in a winery.

The aim of the present work was to develop a sustainable and quantifiable methodology for the determination of HAs, GSM and 2MIB in air using TD-GCMS with application in the analysis of air in a winery. For the development of the proposed methodology, first a TD-GCMS method to analyze the sampling tubs was optimized and validated by applying an experimental design. Then, R-values ($m^3\ h^{-1}$) of each targeted compound were estimated using passive and active sampling. Tenax® TA desorption tubes were selected as the media for adsorption in both sampling techniques and all tubes were analyzed using the validated TD-GCMS method.

## 2. Materials and Methods

### 2.1. Reagents and Solutions

Standards of TCA, TeCA, TBA, PCA, GSM and 2MIB were purchased from Sigma Aldrich (St. Louis, MO, USA). Each compound was dissolved in methanol (gas chromatography grade with >99% purity, supplied by Technochroma) to obtain a concentration of $100\ mg\ L^{-1}$.

### 2.2. Sorbent Tubes

Stainless steel thermal desorption tubes (6 mm O.D. × 90 mm long, 5 mm I.D., Markes International Limited, Pontyclun, UK) were used in this study. Tubes were packed, as described in Batterman et al., 2002, with 200 mg of Tenax® TA supplied by Ingenieria Analitica, S.L (Barcelona, Spain), a porous polymer resin based on 2,6-diphenylene oxide with a particle size of 20–35 mesh, which has been designed for trapping volatile and semi-volatile organic compounds from air. The adsorbed selection was based on the results from Camino-Sanchez et al., 2013 [6]. Tenax® TA tubes showed the highest response for TCA and TCP and the lowest water adsorption capacity. According to the commercial conditions, packed tubes were conditioned at 300 °C for 6 h at 45 mL min$^{-1}$ flow of high purity (99.999%) helium. After conditioning, tubes were sealed with stainless steel caps with Teflon seals screwed onto both ends.

### 2.3. Optimization and Validation of the TD-GCMS Method

In the thermal desorption method, desorption was carried out in two stages. First, the desorption tube was heated inside the desorption unit (at a desorption temperature, TDT) for a certain time (desorption time, DT) using helium as the carrier gas in splitless mode in order to desorbe the analytes and focus them into a cold trap (packed with Tenax TA) which was kept at a chosen temperature (trap low Cryo-temperature, TLCT). Afterwards, the desorption traps were heated to a chosen temperature (or trap high temperature, THT) at maximum heating rate ($50 \,^{\circ}\text{C s}^{-1}$) and the compounds retained were introduced into the chromatographic column. Tube desorption temperature (TDT), desorption time (DT), trap low cryo-temperature (TLCT) and trap high temperature (THT) were optimized together using an experimental design. The Tenax TA tubes, spiked with a $5 \,\text{mg L}^{-1}$ of each analyte, were analysed by GC-MS methodology. The peak area of each compound was the response variable.

An experimental design was used to optimize the TD-GCMS method. For this reason, a procedure to obtain tubes with known concentrations was used. The objective of the optimization was to find the best experimental condition (instrumental values) that maximized the detection of the six compounds simultaneously. A calibration solution loading ring (CSLR) system consists of an unheated injector with a controlled carrier gas supply (nitrogen) and a sorbent tube connection point. The sampling end of the sorbent tube was connected to the CSLR system, and the carrier gas flow rate was set at $80 \,\text{mL min}^{-1}$. The quantity of $1 \,\mu\text{L}$ of a mixed solution, containing all target compounds at concentrations between $0.05 \,\text{mg L}^{-1}$ and $5 \,\text{mg L}^{-1}$ and prepared in methanol, was introduced through the injector septum using a standard GC syringe. Then, a loading time of 3 min was used to ensure that the methanol was eliminated and that the target analytes were retained in sufficient quantity in the tube. In this way, tubes with a known concentration were obtained and could be analyzed by TD-GCMS.

For the analysis of the previously prepared Tenax® TA tubes, an Ultra A automated sampler and a Unity Thermal Desorption system (Markes International Limited, Llanstrisant, UK) connected to a GC-MS was used. First, a desorption and preconcentration phase of the analytes retained on the Tenax® TA tubes was carried out using the thermal desorption system. Second, the detection and quantification was performed using GCMS equipment. The first procedure was optimized by applying an experimental design and the second procedure was validated.

GC-MS analysis was performed with an Agilent 6890 N chromatograph equipped with a Gerstel MPS2 auto-sampler and coupled to an Agilent 5973 N mass spectrometer. The separation was achieved using an HP-5MS column (30 m, 0.25 mm, 0.25 µm film thickness) (J&W Scientific, Folson, CA, USA) and a GC oven program starting at $55 \,^{\circ}\text{C}$ (3 min), increased by $15 \,^{\circ}\text{C min}^{-1}$ to $125 \,^{\circ}\text{C}$, $1.5 \,^{\circ}\text{C min}^{-1}$ to $145 \,^{\circ}\text{C}$, $10 \,^{\circ}\text{C min}^{-1}$ to $183 \,^{\circ}\text{C}$, $1.5 \,^{\circ}\text{C min}^{-1}$ to $195 \,^{\circ}\text{C}$ and $15 \,^{\circ}\text{C min}^{-1}$ to $250 \,^{\circ}\text{C}$ (held for 3 min). The carrier gas was helium (99.999%) from Abello Linde (Barcelona, Spain) with a constant flow rate of $1 \,\text{mL min}^{-1}$. The transfer line temperature was set at $300 \,^{\circ}\text{C}$ and the ion source temperature at $250 \,^{\circ}\text{C}$. The mass spectrometer was operated in selected ion monitoring mode (SIM) detecting the following ion masses: 195 and 210 of TCA, 231 and 246 of TeCA, 346 and 331 of TBA, 280 and 265 of PCA, 112 and 125 of GSM and 95 of 2MIB. GC-MS chromatogram with selected compound peaks and retention times were showed in Figure 1.

The compounds were quantified with calibration response curves generated from five different concentrations.

Instrumental variables (parameters) for the optimization procedure were selected according to (1) targeted compounds, (2) recommended manufacturer values (to improve the analytical assays) and (3) values previously reported in literature (Table 1). In this case, a mixed solution at $5 \,\text{mg L}^{-1}$ was used. The response variable was the area of the chromatographic peak of each compound.

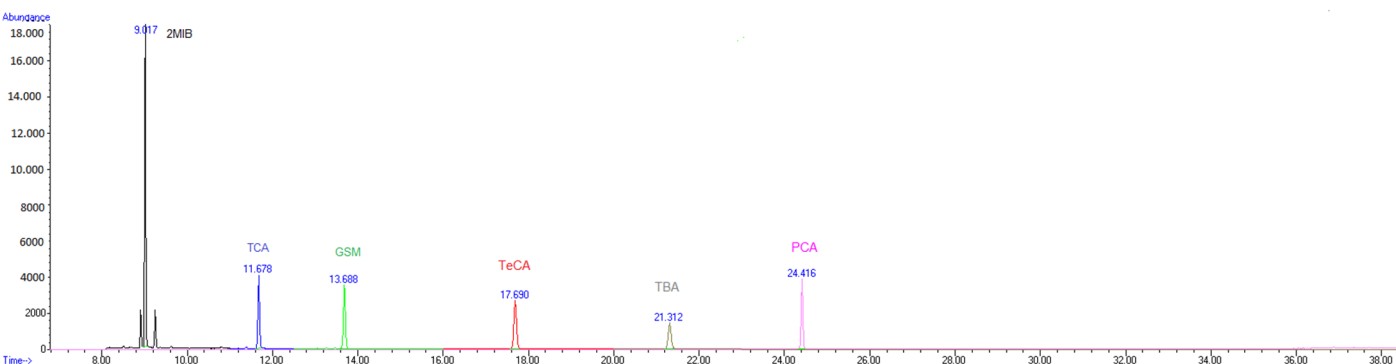

**Figure 1.** Mass spectra and retention time of selected compounds.

**Table 1.** Thermal desorption parameters and values.

| Parameter | Value | Comments |
|---|---|---|
| Desorption time (min) (DT) | 1 to 30 | To be optimized |
| Desorption temperature (°C) (TDT) | 35 to 300 | To be optimized |
| Desorption mode | splitless | Necessary for maximum sensibility |
| Trap purge time (min) | 1.0 | Recommended for Tenax® TA tubes |
| Trap low cryo-temperature (°C) (TLCT) | −20 to 20 | To be optimized |
| Trap heating rate (°C s$^{-1}$) | 50 (MAX) | Previously reported (Camino et al., 2013) |
| Trap high temperature (°C) (THT) | 305 to 320 | To be optimized |

In brief, tubes were heated to 300 °C and the sample was desorbed for 8 min at 40 mL min$^{-1}$. Then, the sample was transferred under a 30 mL min$^{-1}$ helium flow and cryo-focused into the cryotrap at 20 °C. Finally, the cryotrap was rapidly heated to 305 °C to transfer the analytes into the GC column.

The optimization method was also validated in terms of linearity, selectivity, detection and quantification limits, repeatability and reproducibility. A set of six concentrations over an analytical range (from 0.1 to 2 ng tube$^{-1}$) were analysed in triplicate for each compound. The response ratio of the analyte (area) was plotted as a function of analyte concentration and used to generate linear regression. Spiked solutions with low levels of target compounds were used to establish the limit of detection (LOD) and the limit of quantification (LOQ) of the method and overall: LOD is the concentration for signal/noise = 3 and LOQ is the concentration for signal/noise = 10. These limits were manually calculated from the ratio of the peak heights to the average noise before and after each peak. To evaluate the reproducibility and repeatability, spiked tubes at 1 ng tube$^{-1}$ were prepared and analysed on the same day (five replicates) and over a period of 5 days respectively and the results were expressed as %RSD. A blank was analyzed before each analysis.

### 2.4. Determination of R-Values

#### 2.4.1. Passive Sampling Theory

Passive sampling theory was used to calculate the R-values. The fundamental principle of this theory is that chemicals from ambient air accumulate onto the sampling medium by way of gaseous diffusion. The uptake of a gaseous compound from the ambient air to a sampler is described as the effective concentration gradient between the air and the sampler. The mass of analyte retained onto the adsorbent ($M_{ad}$) is a function of the mass

transfer coefficient and the concentration gradient, as defined by the following equation (Camino et al., 2015) Equation (1):

$$-dM_{ad}/dt = k_v \cdot A_s \cdot (C_{air} - C_{ad}/K_{ad}) \tag{1}$$

where $k_v$ is the overall mass transfer coefficient (m s$^{-1}$), $A_s$ is the sampler area (m$^2$), $C_{air}$ is the analyte air concentration (ng m$^3$), $C_{ad}$ is the analyte concentration on the adsorbent (ng m$^3$), and $k_{ad}$ is the sampler/air partition coefficient. For compounds of high $k_{ad}$ and low ambient concentration, the mass transfer from ambient air to the sampling medium of the gaseous compound to be determined is controlled by the air mass transfer rate; therefore, the mass of analyte adsorbed onto the adsorbent is a linear function of time. Analyte concentration in air can be determined by the following equation Equation (2) [13].

$$C_{air} = M_{ad} \cdot (k_v \cdot A_s \cdot t)^{-1} \tag{2}$$

The product of $k_V \times A_S$ is the R-value or sampling rate and can be considered as the main calibration parameter in passive sampling as it allows the easy calculation of the analyte concentration in the air. The R-value may be determined experimentally and is expressed as volume per time (m$^3$ h$^{-1}$). The determination of the R-values has been reviewed elsewhere [13,18,19,24–27]. In the present study, a 10 L chamber was employed to evaluate R-values for different compounds. Therefore, the R-value was calculated using Equation (3), which is derived from Equation (2).

$$M_{ad} = C_{air} (k_v \cdot As) \cdot t \tag{3}$$

According to Equation (3), the mass of each analyte measured in the tube was plotted against time of exposure (t). The slope of each regression is defined as $C_{air} \cdot k_v \cdot A_s$ [13]. Active sampling was employed for monitoring the true concentration of the evaluated compounds inside the calibration chamber ($C_{air}$) and was calculated assuming that all target analytes were completely evaporated.

2.4.2. Experimental Determination of R-Values

R-values (m$^3$ h$^{-1}$) of each targeted compound were calculated using passive and active sampling at the same time. A mid-sized, 10 L glass reactor from Vidrafoc (Barcelona, Spain) was employed as a pollution chamber. At the top of the chamber, 6 mm diameter sampler places were used to position Tenax® TA tubes. The chamber was located in a room with a controlled temperature (21 °C ± 2 °C) but humidity and temperature inside the chamber were also checked during assays using a thermo-hygrometer with a probe (TESTO 635).

Before spiking the chamber, a blank was prepared using active sampling at 140 mL min$^{-1}$ for 15 min to confirm the absence of previous contamination in the chamber air. A handheld air sampling pump SKC model 224-PCMTX8 (Arelco, Fontenay sous Bois, France) was employed to pump air samples through the Tenax® TA tubes. The pump was calibrated using a DFC-HR digital flow meter (Altech, Deerfield, IL, USA) before each sampling. Then, the Tenax® TA tubes were placed in the sampler places and the air in the chamber was subsequently spiked with a solution of 5 mg L$^{-1}$ of each compound. Passive sampling tubes were taken out of the chamber at different times: 1, 6, 24, 30, 48, 54 and 72 h. Active sampling was carried out after collecting each passive tube for monitoring the air concentration of the evaluated compound inside the chamber. As in the case of blanks, active sampling was performed at 140 mL min$^{-1}$ for 15 min using a pump. All tubes were enclosed and stored at 4 °C until they were analysed using the TD-GC-MS procedure previously reported. Finally, the chamber was cleaned with acetone and distilled water and left for 24 h until the next test. This procedure was repeated in triplicate for each compound.

### 2.5. Data Analysis

2.5.1. Optimization

The selected experimental design used for optimizing the TD-GCMS method was a full factorial design with four factors: desorption time (DT) with four levels (1, 8, 15 and 30 min); desorption temperature (TDT) with two levels (35 and 300 °C); trap low cryo-temperature (TLCT) with two levels (−20 and 20 °C); and trap high temperature (THT) with two levels (305 and 320 °C). The selected full factorial design had 32 unique factor level combinations that were replicated twice, leading to a total of 64 experimental runs. The full factorial design allowed estimation of main effects of each of the factors as well as all interactions without confounding.

Plots of the main effects and two-way interactions were inspected and Pareto plots and ANOVA tables were analyzed to assess whether each factor was significantly associated with each response. A separate regression model was fitted to each response (six models in total) including main effects and two-way interactions. Then, each model was simplified using backward elimination so that it included only significant terms. The significance level was set to $\alpha = 0.1$ because we choose to have a higher risk (10%) of concluding that a factor was significant even if no actual difference existed due to the fact that factors are very easy to change. For each model the goodness-of-fit of the regression and the assumptions of the ordinary least squares were checked.

Scatter plots of each pair of response variables and their correlations were inspected to assess the relationship between them. Finally, a response optimization procedure was applied to identify the combination of factor settings that optimized (maximized) the six response variables. The optimization was performed giving the same weight to each factor and by taking into account the best model fit for each response variable. The composite desirability was used to select the best combination of factors. Once the final combination of factors was selected, predicted estimates and their confidence intervals were inspected to determine the range of likely values for a future experiment under those conditions.

Minitab® 18.1 Statistical Software (Addlink Software Cientifico, S.L.) was used to generate and analyze the experimental design. The response variables were the areas of the chromatographic peaks of each compound that were obtained employing ChemStation version 3.1 software.

2.5.2. Determination of R-Values

R-values ($m^3 \ h^{-1}$) were experimentally calculated for each compound by the creation of a linear plot of the mass of analyte ($M_{ad}$, ng) against time (h) at a selected initial air concentration (5 mg $L^{-1}$) of the studied compound according to Equation (3). $M_{ad}$ was obtained for each selected time using passive sampling. The slope of each regression is defined as $C_{air} \cdot k_v \cdot A_s$, the R-value for an analyte being $k_v \cdot A_s$ [13] and $C_{air}$ the average concentration of this analyte in the air (ng $m^{-3}$). The latter was calculated by active sampling. Therefore, R-values were calculated by dividing the obtained slope by $C_{air}$.

Validation and R-values measurements were performed using Microsoft Excel 2015.

## 3. Results

### 3.1. Optimization of Thermal Desorption Method (TD)

Table 2 summarizes the significant associations between the factors and each of the six compounds. In all cases, factors DT and TDT are significantly positively associated with each compound, i.e., increasing the factor increases the response of the compound (see Figures A1, A3, A5, A7, A9 and A11 in Appendix A). The interaction between DT and TDT is significant in compounds TCA, GMN and TeCA (see Figures A4, A6 and A8) and the interaction TLCT and THT is significant in TCA, TBA and PCA (see Figures A4, A10 and A12). When the latter interaction is significant, the main effects TLCT and THT also need to be in the model even though they are not significant (we have not reported them in Table 2 for simplicity).

**Table 2.** Factors associated with each of the six compounds. A + (−) indicates that there is a positive (negative) association, i.e., increasing the factor increases (decreases) the value of the response.

| | Main Effects | | | | Interactions | |
|---|---|---|---|---|---|---|
| Response | DT | TDT | TLCT | THT | DT*TDT | TLCT*THT |
| MIB | + | + | NA | NA | NA | NA |
| TCA | + | + | NA | NA | Yes | Yes |
| GSM | + | + | NA | NA | Yes | NA |
| TeCA | + | + | NA | NA | Yes | NA |
| TBA | + | + | + | − | NA | Yes |
| PCA | + | + | + | − | NA | Yes |

NA: not applicable.

Plots of main effects and interactions (Figures A1–A12), as well as the model coefficients (Tables A1–A6) together with the regression equation, are shown in Appendix A to enhance interpretability of the individual models. Results of the 62 experiments are in Appendix B.

In all models, the assumptions of the ordinary least squares are met except for two runs that show an anomalously low value in all compounds. These are runs 16 and 63 (see Table 7). When compared with their replicates (runs 15 and 64, respectively, from Table 7), much lower quantities of those compounds were found on those runs, which makes the residuals very low. The lack of fit test is non-significant on 2MIB, GSM and TeCA. For the other compounds, the lack of fit test is significant, showing that the model does not fit the data well, mainly due to the large variability between some replicates such as in runs 16 and 63.

For all compounds, the greater response is achieved with TDT = 300 and DT = 30 as can be seen in all interaction plots (DT*TDT): in most cases the response is more than double for the same DT value when TDT is set to 300 instead of 35. These results would relate to the volatility of the compounds. The interaction between TLCT and THT (TLCT*THT), even if significant, has a much lower effect size: in all cases, when THT is at 305, increasing TLCT increases the response while, when THT is at 320, increasing TLCT reduces the response.

Overall, the experimental conditions that give higher responses in one compound also give higher responses in the other compounds. Table 3 shows the Pearson correlation coefficients between each pair of responses; it can be seen that all values are greater than 0.776 which indicates that those experimental conditions that detect greater values of one compound also have a greater detection of the other compounds. This will facilitate the optimization of the six compounds simultaneously.

The optimization procedure to find the best combination of factors to jointly maximize the six compounds (with equal weights and importance) showed that the optimal solution was: DT = 30, TDT = 300, TLCT = 20 and THT = 305. This combination gave a composite desirability of 0.66. The predicted responses (fit) for this combination are shown on Table 4. Note that none of the 95% confidence intervals were below 91,872.

**Table 3.** Pearson correlation coefficients between the measurements of the six compounds under the 64 experimental conditions.

| | 2MIB | TCA | GSM | TeCA | TBA |
|---|---|---|---|---|---|
| TCA | 0.776 | | | | |
| GSM | 0.899 | 0.875 | | | |
| TeCA | 0.848 | 0.937 | 0.972 | | |
| TBA | 0.816 | 0.889 | 0.916 | 0.952 | |
| PCA | 0.790 | 0.865 | 0.868 | 0.906 | 0.953 |

**Table 4.** Predicted response variables (and 95% CI) for the six compounds under two experimental conditions. In all cases TDT = 300, TLCT = 20 and THT = 305. The optimal setting is with DT = 30; DT = 8 is the optimal setting for a desorption time of 8 min.

| | Predicted Response (95% CI) | |
|---|---|---|
| | **DT = 30** | **DT = 8** |
| PCA | 248,455 (122,899; 374,011) | 177,353 (135,501; 219,205) |
| TBA | 194,295 (103,912; 284,678) | 120,356 (90,228; 150,484) |
| TeCa | 205,863 (91,872; 319,854) | 183,980 (145,983; 221,977) |
| GSM | 417,875 (165,319; 670,431) | 417,603 (333,418; 501,788) |
| TCA | 266,963 (128,930; 404,996) | 252,412 (199,550; 305,275) |
| 2MIB | 407,913 (162,156; 653,670) | 351,452 (285,296; 417,607) |

However, because the DT is the critical value in operating conditions due to the need to analyze a larger number of samples in the minimum amount of time, the optimization was repeated, fixing DT at 8 and the optimal conditions were DT = 8, TDT = 300, TLCT = 20 and THT = 305. Under this scenario, the composite desirability was 0.55 (predicted responses are shown on Table 4). Under this experimental condition none of the compounds had a lower 95% confidence interval below 90,228.

*3.2. TD-GC-MS Method Validation*

The GC-MS method used to analyze all Tenax® TA tubes was validated using the parameters DT = 8, TDT = 300, TLCT = 20 and THT = 305. The linearity of the method was studied in the range of 0.1 to 2 ng tube$^{-1}$ at six concentration levels. As shown in Table 5, multiple $R^2$ coefficients ($R^2$) were higher than 0.994. LOD and LOQ values are shown in Table 5 and were determined visually, applying a signal-to-noise ratio established at 3 and 10 respectively. The limits of quantification were from 0.01 to 0.06 ng tube$^{-1}$. These results are generally similar to those reported in other studies of thermal desorption that use TD-GCMS [28,29]. The precision (reproducibility and repeatability) of the methodology was assessed expressing the random error of a set of individual measurements by means of the relative standard deviation (%RSD). As can be seen in Table 5, RSD (at 1 ng tube$^{-1}$) values were low, ranging from 2.5% to 9.3% for repeatability and from 3.9% and 9.9% for reproducibility. These values are similar to others reported by Camino et al., 2015 [13] for TCA and TCP.

**Table 5.** Results of the method validation (linearity range from 0.1–2 ng tube$^{-1}$): linearity ($R^2$), limits of detection (LOD) and quantification (LOQ), repeatability and reproducibility (expressed as %RSD) of the TD-GCMS method.

| Target Compounds | Quantification Ions | | $R^2$ | LOD (ng tube$^{-1}$) | LOQ (ng tube$^{-1}$) | Repeatability (%RSD at 1 ng tube$^{-1}$) [a] | Reproducibility (%RSD at 1 ng tube$^{-1}$) [a] |
|---|---|---|---|---|---|---|---|
| | *m/z* 1 | *m/z* 2 | | | | | |
| TCA | 195 | 210 | 0.995 | 0.01 | 0.05 | 9.3 | 9.9 |
| TeCA | 231 | 246 | 0.994 | 0.06 | 0.1 | 6.0 | 9.6 |
| TBA | 346 | 331 | 0.995 | 0.05 | 0.1 | 6.8 | 5.7 |
| PCA | 280 | 265 | 0.994 | 0.03 | 0.05 | 3.5 | 3.9 |
| GSM | 112 | 125 | 0.995 | 0.03 | 0.05 | 8.9 | 9.0 |
| 2MIB | 95 | - [b] | 0.994 | 0.03 | 0.06 | 2.5 | 3.9 |

[a] RSD: relative standard deviation (*n* = 5). [b] None.

### 3.3. R-Values

Table 6 summarizes the results of the passive sampling at each testing time and the R-values at $20 \pm 3\,°C$ and $21 \pm 2\%$ of humidity. According to our results, all compounds were detected from the first (2 h after the start of the assay) to the last sampling (96 h). The behaviour of all analytes was the same: increasing over exposure time until it decreased. Therefore, the tubes used for the determination of R-values were those from when it was increasing. The average R-values of each compound together with the RSD value is also presented in Table 6.

**Table 6.** Mass of TCA, TeCA, TBA, PCA, GSM and 2MIB (Mad, ng) at different exposure times. R-values obtained at $20 \pm 3\,°C$ and $21 \pm 2\%$ of humidity.

| Exposure Time (h) | $M_{ad}$, ng (Mean $\pm$ Standard Deviation) | | | | | |
|---|---|---|---|---|---|---|
| | TCA | TeCA | TBA | PCA | GSM | 2MIB |
| 2 | $0.3 \pm 0.1$ | $1.2 \pm 0.20$ | $1.8 \pm 0.0$ | $1.3 \pm 1.1$ | $0.8 \pm 0.3$ | $0.1 \pm 0.0$ |
| 6 | $1.2 \pm 0.8$ | $3.0 \pm 0.42$ | $3.9 \pm 2.5$ | $2.7 \pm 0.0$ | $0.9 \pm 0.2$ | $0.7 \pm 0.0$ |
| 24 | $1.7 \pm 0.6$ | $6.32 \pm 0.01$ | $17.6 \pm 0.0$ | $13.2 \pm 0.8$ | $3.4 \pm 1.2$ | $1.3 \pm 0.1$ |
| 48 | $2.6 \pm 1.1$ | $10.12 \pm 0.01$ | $20.1 \pm 0.0$ | $23.4 \pm 0.5$ | $5.5 \pm 2.0$ | $1.8 \pm 0.0$ |
| 54 | $2.7 \pm 0.8$ | $11.86 \pm 0.0$ | $17.4 \pm 0.0$ | $20.9 \pm 0.2$ | $6.9 \pm 0.1$ | $2.1 \pm 0.1$ |
| 72 | $2.8 \pm 0.1$ | $15.48 \pm 0.13$ | $16.6 \pm 0.6$ | $20.4 \pm 0.7$ | $9.3 \pm 2.2$ | $2.2 \pm 0.1$ |
| 96 | $3.2 \pm 0.09$ | $14.00 \pm 0.0$ | $13.4 \pm 0.0$ | $18.1 \pm 0.5$ | $12.8 \pm 3.2$ | $0.9 \pm 0.1$ |
| 120 | $4.1 \pm 0.6$ | $12.2 \pm 0.0$ | $8.0 \pm 0.0$ | $16.2 \pm 0.1$ | $7.2 \pm 1.2$ | $0.8 \pm 0.2$ |
| R-values ($m^3\ h^{-1}$) | $0.035 \pm 17.1$ | $0.028 \pm 0.8$ | $0.013 \pm 26.2$ | $0.071.3 \pm 3.9$ | $0.032.3 \pm 13.4$ | $0.025 \pm 28.8$ |

The response of these tubes (mainly from 2 to 54 or 72 h) showed good linearity with a coefficient of determination ($R^2$) between 0.996 to 0.999 for all compounds. The R-values calculated from the correlation graph's slope and average $C_{air}$ were from 0.013 to $0.071\ m^3\ h^{-1}$. An example of a calibration curve obtained for calculating the R-values (in the case of TCA) is presented in Figure 2.

The diffusive uptake rates for airborne volatile organic compounds (VOCs) have been determined by implementing empirical models by Jia and Fu 2017 [17]. In their case they also used thermal desorption techniques and tubes packed with Tenax TA. Although the diffusive sampling rate can be dependent on certain factors such as the compound or time of exposure [30], the results obtained have the same order of magnitude as ours (0.078 from $0.0354\ m^3\ h^{-1}$). An R-value of $0.080\ m^3\ h^{-1}$ for polychlorinated biphenil was also obtained by Persoon and Hornbuckle (2009) [31]. In the case of TCA, our result ($0.035\ m^3\ h^{-1}$) is higher than the value previously described ($0.0019\ m^3\ h^{-1}$) [13].

### 3.4. Application of the Method in Wineries

In order to evaluate the performance of the proposed analytical method in real air samples, the air at two wineries was monitored using passive sampling. We placed the sampling tubes near the production area (w1), in the bottle cellar (w2) and storage (w3) in the case of Winery 1 and near to the production area (w4) in Winery 2. Sampling points were located in the middle of the area being sampled and it was ensured that the air was not coming directly from outdoors. The assays were conducted in duplicate. The minimum sampling time depended on the concentration of the analytes in the air because it had to be long enough to reach a mass of analyte in the sorbent and higher than the minimum measurable amount [13]. For this reason, five exposure times were selected (24, 72, 120, 168 h). The results or concentrations of selected compounds correspond to the mean of the concentrations obtained at each selected time. In addition, the temperature was checked but not the humidity as it would seem that this is not relevant during sampling [13]. Further, $0.008\ ng\ m^{-3}$ of TCA and $0.003\ ng\ m^{-3}$ of PCA were found in the bottle cellar of winery 1

(Table 7). Both compounds were detected in all selected periods. Neither TecA, TBA, GSM and 2MIB were found. None of the selected compounds were observed in Winery 2 (w2, w3 and w4).

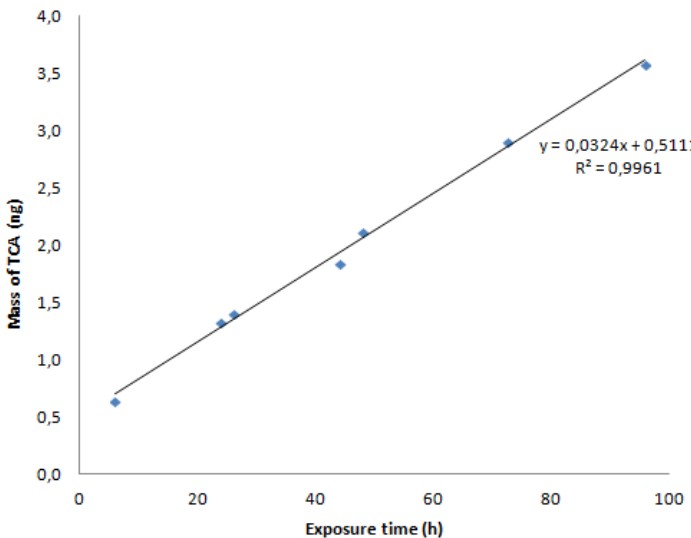

**Figure 2.** Calibration curve for TCA.

**Table 7.** Concentrations of TCA, TeCA, TBA, PCA, GSM and 2MIB (ng·m$^{-3}$) found in wineries and cork facilities. Mean $\pm$ standard deviation.

|      | TCA               | TeCA | TBA         | PCA               | GSM         | 2MIB        |
| ---- | ----------------- | ---- | ----------- | ----------------- | ----------- | ----------- |
| w1   | 0.008 $\pm$ 0.006 | <LD  | <LD         | 0.003 $\pm$ 0.003 | <LD         | <LD         |
| w2   | <LD               | <LD  | <LD         | <LD               | <LD         | <LD         |
| w3   | <LD               | <LD  | <LD         | <LD               | <LD         | <LD         |
| w4   | <LD               | <LD  | <LD         | <LD               | <LD         | <LD         |

LD is the limit of detection (see Table 5).

A higher TCA concentration ($72 \pm 13$ ng m$^{-3}$) was found by Camino et al., 2015 [13] during the analysis of the air in a cellar.

## 4. Discussion

A sustainable and quantifiable methodology for the determination of HAs, GSM and 2MIB in air using TD-GCMS was developed. This methodology allows the detection and quantification of the selected compounds in air using a passive sampling procedure.

The experimental results showed that DT and TDT are significantly associated with all compounds: the higher the DT and TDT values are, the higher the detection of the compounds. The interaction between DT and TDT is also significant on TCA, GMN and TeCA, and in all three compounds it can be interpreted as: the compound detection (response) increases when DT increases (e.g., moving from 1 to 8), but this effect is higher when TDT is at 300 than when TDT is at 35. This is because, by increasing the desorption time and temperature, the adsorbed compounds are released in greater quantities due to their volatilization as the temperature increases. The final result is greater quantities of the selected compounds in the GC-MS inlet.

The other significant interaction was between TLCT and THT on compounds TCA, TBA and PCA. In each of these cases, the interpretation of the interaction is as follows: when THT is at 305, increasing TLCT increases the response; while when THT is at 320, increasing TLCT reduces the response. This finding does not correspond to what we would expect because lower TLCT values should increase the preconcentration of the compounds on the trap and higher THT values would increase the subsequent release of

these compounds. The final result should be an increase in their amounts in the GC-MS inlet and therefore a greater response. The Type I error (false positive) for the experimental design was set to 0.1 and, particularly in the case of this interaction, the risk of determining that the interaction is significant while it is, in fact, not, is of 0.061, 0.077 and 0.060 for TCA, TBA and PCA, respectively. Therefore, more assays at different initial concentrations of the selected compounds should be carried out in subsequent studies

The experimental results also showed that all compounds could be maximized together because the correlation between them was high (greater than 0.766). The conditions that jointly maximize the six compounds were DT = 30, TDT = 300, TLCT = 20 and THT = 305 giving a composite desirability of 0.66. Those conditions lead to results similar to reducing the desorption time to 8 min (DT = 8, TDT = 300, TLCT = 20 and THT = 305). This is an important finding, allowing the samples to be analyzed in a shorter amount of time.

The optimized method displays good linearity over the concentration ranges explored, as well as good repeatability and reproducibility (both with RSD below 10%). In addition, sampling rates of selected compounds were experimentally calculated and the obtained values ranged from 0.013 to 0.071 $m^3 \ h^{-1}$. The applicability of the technique developed in real samples was tested in different places in two wineries. The method showed enough sensitivity to detect TCA and PCA within the selected periods. In conclusion, the developed methodology can be used for the monitoring and quantification of all selected compounds in air using passive sampling.

## 5. Conclusions

A sustainable and quantifiable methodology for the determination of HAs, GSM and 2MIB in air using TD-GCMS with application in the analysis of air in a winery was developed. In brief, tubes were heated to 300 °C and the sample was desorbed for 8 min at 40 mL $min^{-1}$. Then, the sample was transferred under a 30 mL $min^{-1}$ helium flow and cryo-focused into the cryotrap at 20 °C. Finally, the cryotrap was rapidly heated to 305 °C to transfer the analytes into the GC column. The developed and optimized method displays good linearity over the concentration ranges explored, as well as good repeatability and reproducibility.

Then, sampling rates or R-values ($m^3 \ h^{-1}$) of each targeted compound were estimated and the obtained values ranged from 0.013 to 0.071 $m^3 \ h^{-1}$. The method showed enough sensitivity to detect TCA and PCA in wineries. The developed methodology can be used for the monitoring and quantification of all selected compounds in air using passive sampling.

**Author Contributions:** Conceptualization, M.V.-M., M.V. and P.J.; methodology, R.D.N., M.V.-M. and P.J.; software, M.V.-M. and P.J.; validation, M.V.-M. and P.J.; analysis, R.D.N., M.V.-M. and P.J.; investigation, M.V.-M. and P.J.; resources, M.V.-M., M.V., R.D.N. and P.J.; data curation, M.V.-M. and P.J.; writing—original draft preparation, M.V.-M. and P.J.; writing—review and editing, M.V.-M. and P.J.; visualization, R.D.N., M.V.-M., M.V. and P.J.; supervision, M.V.-M. and P.J.; project administration, M.V. and P.J. All authors have read and agreed to the published version of the manuscript.

**Funding:** This research received no external funding.

**Institutional Review Board Statement:** Not applicable.

**Informed Consent Statement:** Not applicable.

**Acknowledgments:** The authors are grateful to Júlia Plaja for her collaboration in laboratory tasks.

**Conflicts of Interest:** The authors declare no conflict of interest.

## Appendix A

For each compound, the plots of the main effects and the two-way interaction are represented in order to better understand the individual model results. In each case, the gray background represents a term that is not included (not significant) in the final retained model. Moreover, a table with the model coefficients, as well as their standard

deviation and the *p*-value, is shown for each compound. Finally, the regression equation is represented.

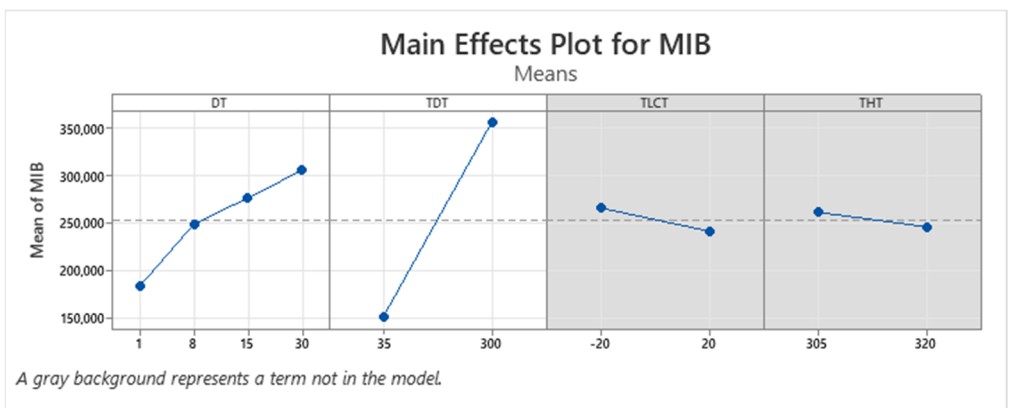

**Figure A1.** Main effects plot for MIB model.

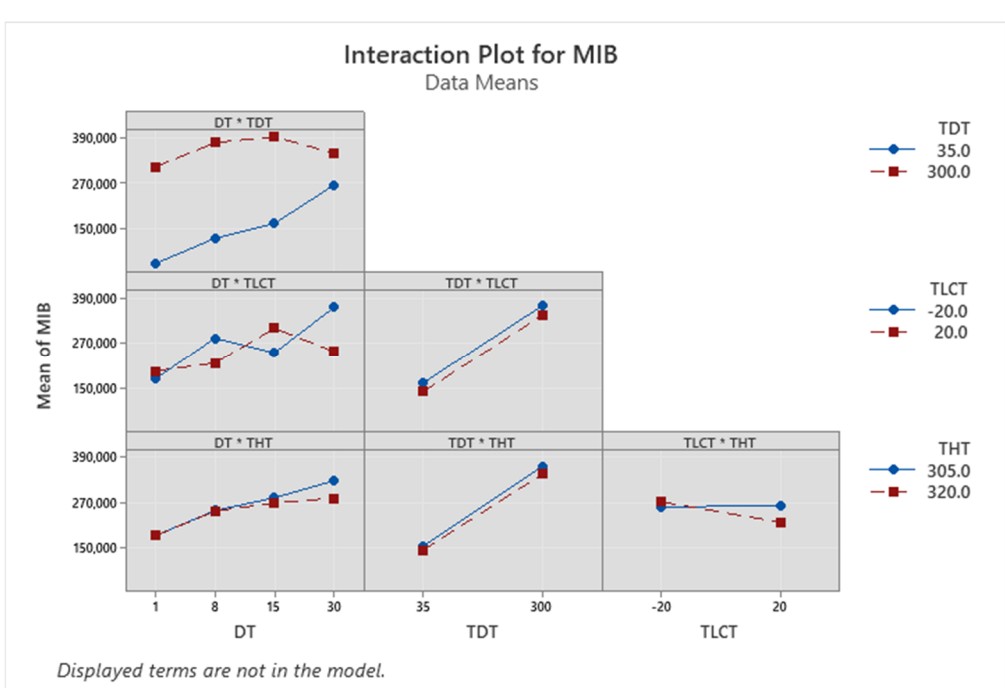

**Figure A2.** Interaction plots for MIB model.

**Table A1.** Model coefficients for MIB.

| Term | Coef | SE Coef | *p*-Value |
|---|---|---|---|
| Constant | 253,509 | 14,785 | 0.000 |
| DT = 1 | −69,950 | 25,609 | 0.008 |
| DT = 8 | −4656 | 25,609 | 0.856 |
| DT = 15 | 22,802 | 25,609 | 0.377 |
| TDT = 35 | −102,599 | 14,785 | 0.000 |

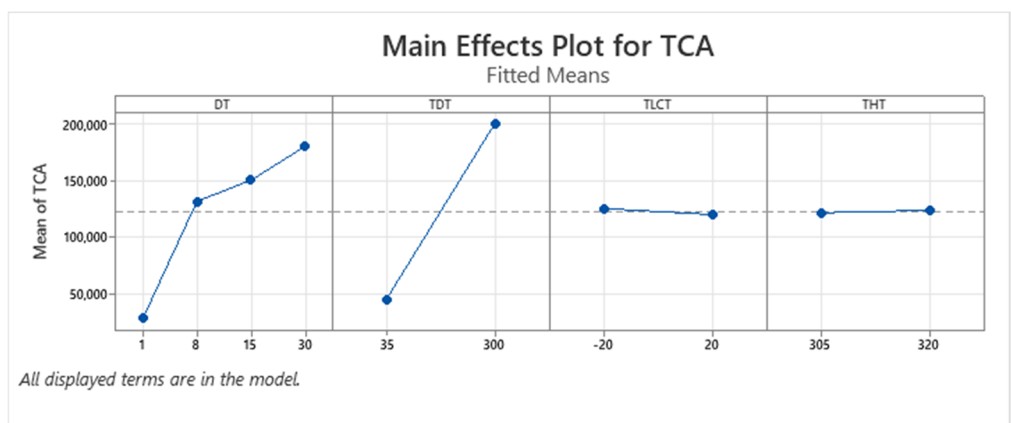

**Figure A3.** Main effects plot for TCA model.

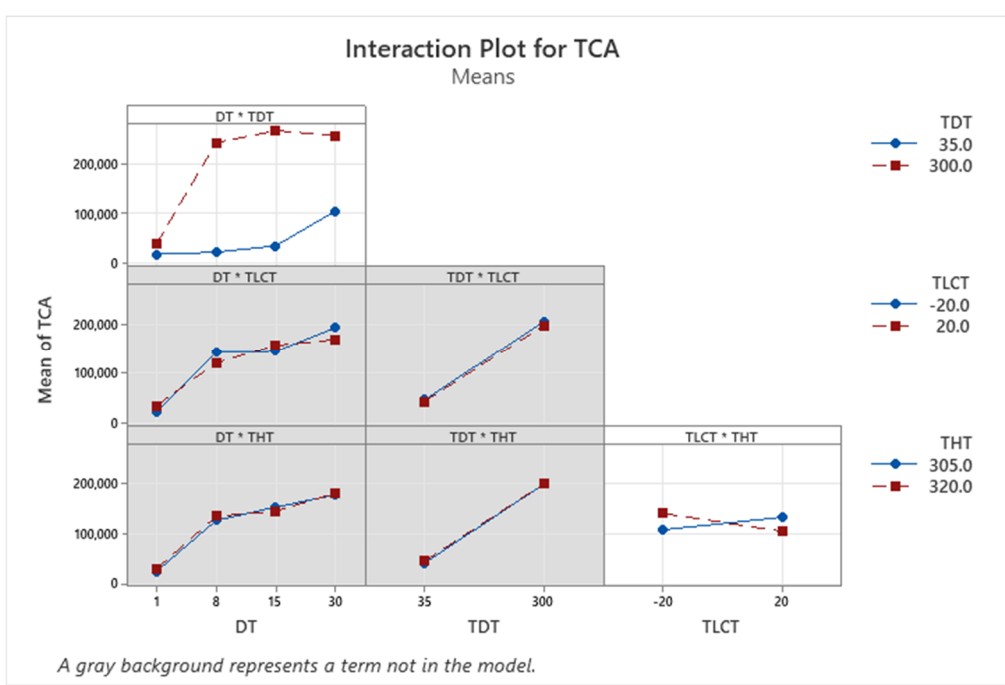

**Figure A4.** Interaction plots for TCA model.

**Table A2.** Model coefficients for TCA.

| Term | Coef | SE Coef | *p*-Value |
| --- | --- | --- | --- |
| Constant | 122,107 | 7947 | 0.000 |
| DT = 1 | −94,342 | 13,764 | 0.000 |
| DT = 8 | 9002 | 13,764 | 0.516 |
| DT = 15 | 27,766 | 13,764 | 0.049 |
| TDT = 35 | −77,931 | 7947 | 0.000 |
| TLCT = -20 | 2701 | 7947 | 0.735 |
| THT = 305 | −1109 | 7947 | 0.889 |
| DT = 1 TDT = 35 | 67,602 | 13,764 | 0.000 |
| DT = 8 TDT = 35 | −31,962 | 13,764 | 0.024 |
| DT = 15 TDT = 35 | −37,698 | 13,764 | 0.008 |
| TLCT = -20 THT = 305 | −15,219 | 7947 | 0.061 |

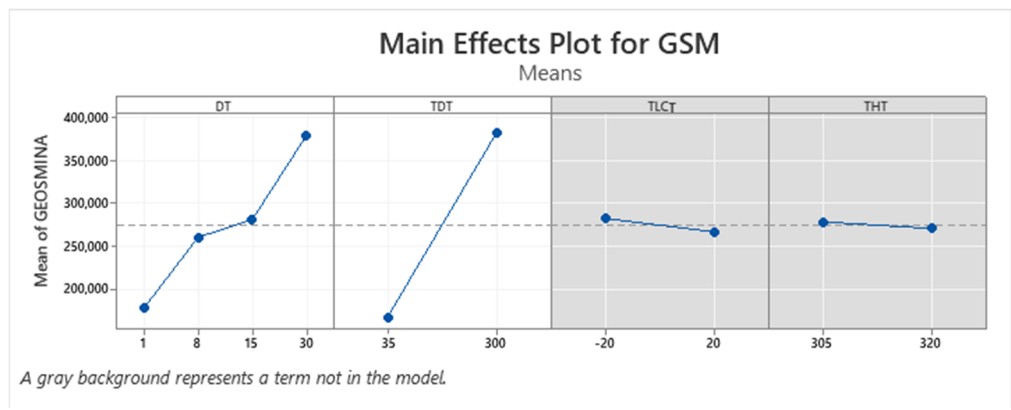

**Figure A5.** Main effects plot for GSM model.

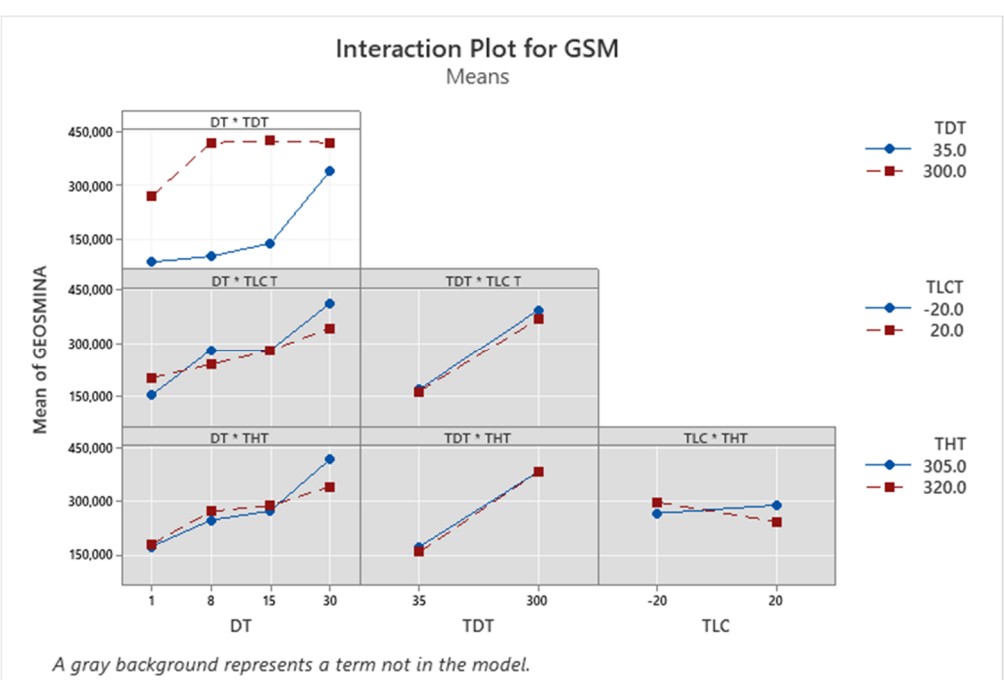

**Figure A6.** Interaction plots for GSM model.

**Table A3.** Model coefficients for GSM.

| Term | Coef | SE Coef | *p*-Value |
|---|---|---|---|
| Constant | 274,084 | 14,858 | 0.000 |
| DT = 1 | −96,736 | 25,735 | 0.000 |
| DT = 8 | −14,084 | 25,735 | 0.586 |
| DT = 15 | 6418 | 25,735 | 0.804 |
| TDT = 35 | −107,638 | 14,858 | 0.000 |
| DT = 1 TDT = 35 | 16,419 | 25,735 | 0.526 |
| DT = 8 TDT = 35 | −49,966 | 25,735 | 0.057 |
| DT = 15 TDT = 35 | −34,701 | 25,735 | 0.183 |

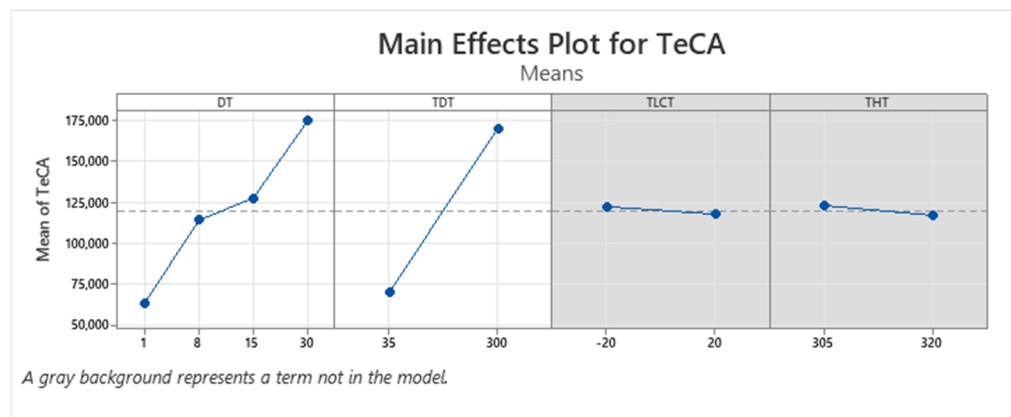

**Figure A7.** Main effects plot for TeCA model.

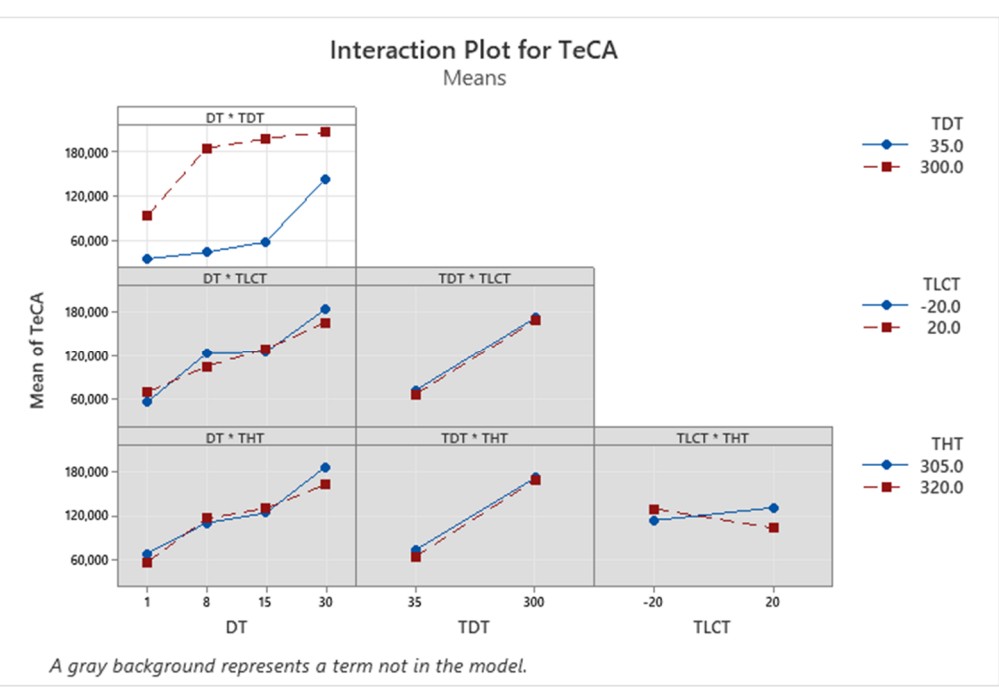

**Figure A8.** Interaction plots for TeCA model.

**Table A4.** Model coefficients for TeCA.

| Term | Coef | SE Coef | *p*-Value |
|---|---|---|---|
| Constant | 119,559 | 6706 | 0.000 |
| DT = 1 | −56,416 | 11,615 | 0.000 |
| DT = 8 | −5993 | 11,615 | 0.608 |
| DT = 15 | 7603 | 11,615 | 0.515 |
| TDT = 35 | −50,143 | 6706 | 0.000 |
| DT = 1 TDT = 35 | 21,250 | 11,615 | 0.073 |
| DT = 8 TDT = 35 | −20,271 | 11,615 | 0.086 |
| DT = 15 TDT = 35 | −19,625 | 11615 | 0.097 |

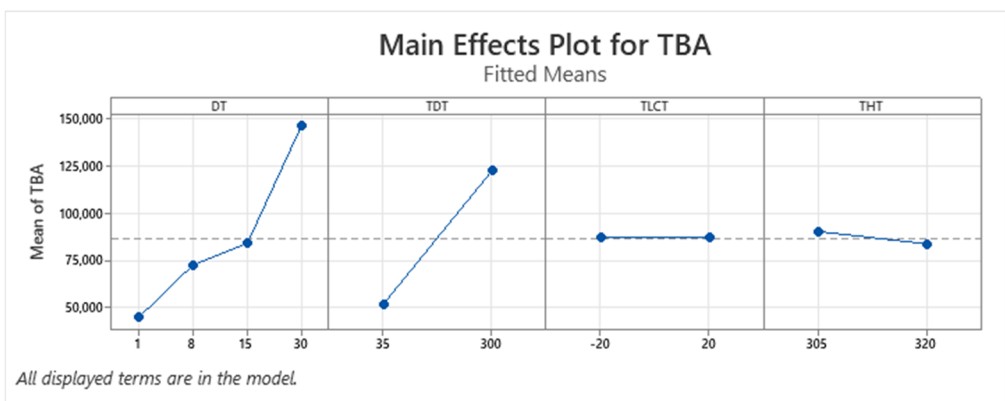

**Figure A9.** Main effects plot for TBA model.

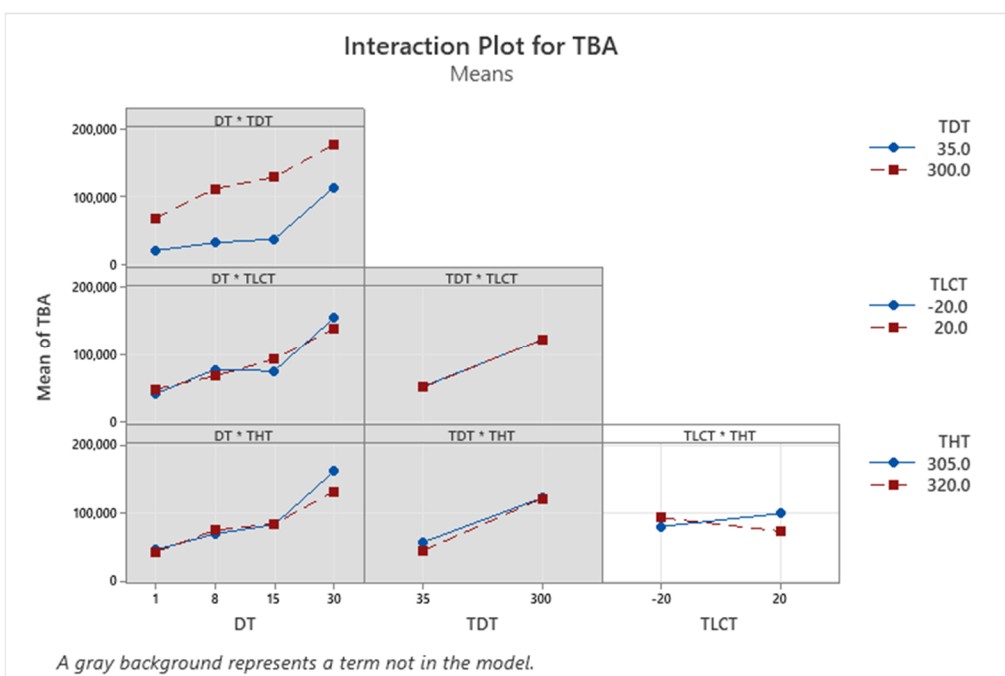

**Figure A10.** Interaction plots for TBA model.

**Table A5.** Model coefficients for TBA.

| Term | Coef | SE Coef | *p*-Value |
|---|---|---|---|
| Constant | 86,714 | 5317 | 0.000 |
| DT = 1 | −42,159 | 9210 | 0.000 |
| DT = 8 | −14,411 | 9210 | 0.123 |
| DT = 15 | −2959 | 9210 | 0.749 |
| TDT = 35 | −35,341 | 5317 | 0.000 |
| TLCT = −20 | 124 | 5317 | 0.981 |
| THT = 305 | 3251 | 5317 | 0.543 |
| TLCT = −20 THT = 305 | −9584 | 5317 | 0.077 |

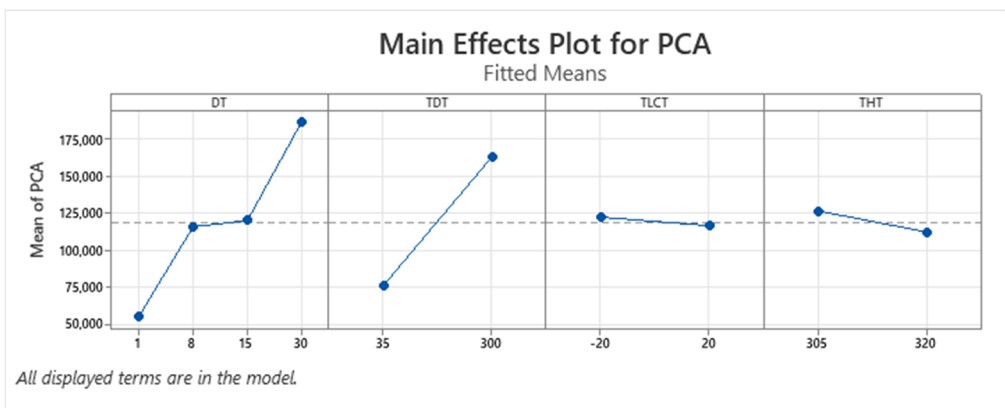

**Figure A11.** Main effects plot for PCA model.

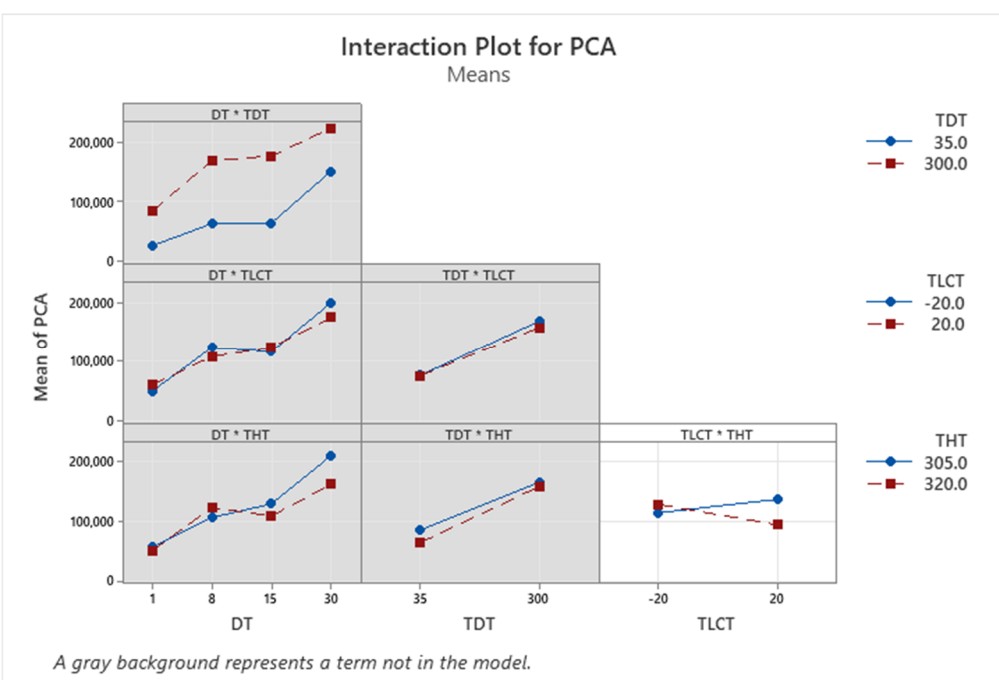

**Figure A12.** Interaction plots for PCA model.

**Table A6.** Model coefficients for PCA.

| Term | Coef | SE Coef | *p*-Value |
|---|---|---|---|
| Constant | 118,958 | 7386 | 0.000 |
| DT = 1 | −64,400 | 12,794 | 0.000 |
| DT = 8 | −3661 | 12,794 | 0.776 |
| DT = 15 | 620 | 12,794 | 0.962 |
| TDT = 35 | −43,603 | 7386 | 0.000 |
| TLCT = −20 | 2866 | 7386 | 0.699 |
| THT = 305 | 7159 | 7386 | 0.337 |
| TLCT = −20 THT = 305 | −14,159 | 7386 | 0.060 |

## Appendix B

Results of the full factorial design with four factors (32 unique factor levels with two replicates each): desorption time (DT) with four levels (1, 8, 15 and 30 min); desorption temperature (TDT) with two levels (35 and 300 °C); trap low cryo-temperature (TLCT) with two levels (−20 and 20 °C); and trap high temperature (THT) with two levels (305 and 320 °C).

**Table 7.** Results of the design of experiments.

| Run (Standard Order) | DT | TDT | TLCT | THT | MIB | TCA | GSM | TeCA | TBA | PCA |
|---|---|---|---|---|---|---|---|---|---|---|
| 1 | 1 | 35 | −20 | 305 | 39,729 | 5980 | 51,859 | 30,126 | 24,167 | 29,759 |
| 2 | 1 | 35 | −20 | 305 | 41,691 | 2051 | 26,508 | 37,057 | 19,137 | 41,719 |
| 3 | 8 | 35 | −20 | 305 | 1062 | 5329 | 2128 | 7549 | 4438 | 8390 |
| 4 | 8 | 35 | −20 | 305 | 113,150 | 27,479 | 123,655 | 67,597 | 45,365 | 79,090 |
| 5 | 15 | 35 | −20 | 305 | 287,128 | 73,984 | 282,752 | 122,206 | 68,384 | 108,877 |
| 6 | 15 | 35 | −20 | 305 | 29,209 | 13,847 | 38,480 | 15,355 | 9794 | 18,525 |
| 7 | 30 | 35 | −20 | 305 | 356,711 | 102,751 | 443,958 | 177,078 | 152,746 | 207,312 |
| 8 | 30 | 35 | −20 | 305 | 528,734 | 130,252 | 480,067 | 167,733 | 140,451 | 205,729 |
| 9 | 1 | 300 | −20 | 305 | 321,284 | 31,773 | 249,483 | 74,043 | 73,626 | 101,608 |
| 10 | 1 | 300 | −20 | 305 | 223,313 | 26,527 | 198,734 | 75,342 | 47,031 | 46,104 |
| 11 | 8 | 300 | −20 | 305 | 506,839 | 291,312 | 519,946 | 225,261 | 139,775 | 208,774 |
| 12 | 8 | 300 | −20 | 305 | 410,427 | 241,360 | 429,038 | 184,862 | 111,428 | 162,671 |
| 13 | 15 | 300 | −20 | 305 | 352,257 | 245,729 | 418,726 | 191,182 | 114,042 | 176,334 |
| 14 | 15 | 300 | −20 | 305 | 457,017 | 253,548 | 470,958 | 190,315 | 115,926 | 165,099 |
| 15 | 30 | 300 | −20 | 305 | 430,410 | 276,330 | 441,079 | 219,637 | 184,751 | 229,804 |
| 16 | 30 | 300 | −20 | 305 | 41,811 | 7421 | 88,853 | 38,062 | 37,029 | 47,382 |
| 17 | 1 | 35 | 20 | 305 | 12,286 | 6455 | 17,981 | 3676 | 800 | 1225 |
| 18 | 1 | 35 | 20 | 305 | 57,049 | 10,238 | 97,838 | 48,081 | 39,478 | 53,325 |
| 19 | 8 | 35 | 20 | 305 | 165,375 | 16,977 | 125,683 | 35,766 | 35,003 | 74,158 |
| 20 | 8 | 35 | 20 | 305 | 95,354 | 14,450 | 81,251 | 55,176 | 41,365 | 63,699 |
| 21 | 15 | 35 | 20 | 305 | 256,854 | 31,085 | 168,048 | 55,463 | 48,429 | 88,828 |
| 22 | 15 | 35 | 20 | 305 | 66,263 | 65,907 | 152,733 | 69,075 | 26,870 | 38,613 |
| 23 | 30 | 35 | 20 | 305 | 200,276 | 83,481 | 314,728 | 144,049 | 113,449 | 146,723 |
| 24 | 30 | 35 | 20 | 305 | 239,546 | 86,190 | 372,967 | 143,877 | 149,889 | 211,230 |
| 25 | 1 | 300 | 20 | 305 | 457,141 | 25,281 | 475,253 | 164,570 | 88,351 | 88,197 |
| 26 | 1 | 300 | 20 | 305 | 316,784 | 93,303 | 275,923 | 119,945 | 77,313 | 99,826 |
| 27 | 8 | 300 | 20 | 305 | 371,922 | 216,136 | 350,845 | 151,718 | 85,765 | 117,775 |
| 28 | 8 | 300 | 20 | 305 | 332,791 | 202,924 | 350,717 | 156,955 | 93,797 | 140,004 |
| 29 | 15 | 300 | 20 | 305 | 467,305 | 306,122 | 238,279 | 154,174 | 154,174 | 250,276 |
| 30 | 15 | 300 | 20 | 305 | 350,350 | 240,497 | 416,198 | 194,518 | 127,276 | 190,512 |
| 31 | 30 | 300 | 20 | 305 | 446,550 | 401,763 | 645,897 | 325,578 | 275,582 | 346,003 |
| 32 | 30 | 300 | 20 | 305 | 380,842 | 335,447 | 542,584 | 274,466 | 233,260 | 288,172 |
| 33 | 1 | 35 | −20 | 320 | 34,947 | 6040 | 51,342 | 28,349 | 19,429 | 20,902 |
| 34 | 1 | 35 | −20 | 320 | 74,483 | 30,764 | 147,709 | 52,043 | 28,349 | 23,773 |
| 35 | 8 | 35 | −20 | 320 | 263,973 | 38,899 | 187,344 | 82,755 | 51,849 | 87,776 |

**Table 7.** *Cont.*

| Run (Standard Order) | DT | TDT | TLCT | THT | MIB | TCA | GSM | TeCA | TBA | PCA |
|---|---|---|---|---|---|---|---|---|---|---|
| 36 | 8 | 35 | −20 | 320 | 135,191 | 13,940 | 75,521 | 21,590 | 27,166 | 69,227 |
| 37 | 15 | 35 | −20 | 320 | 6711 | 2025 | 2963 | 1651 | 1240 | 2013 |
| 38 | 15 | 35 | −20 | 320 | 177,200 | 33,002 | 156,131 | 68,055 | 41,884 | 71,770 |
| 39 | 30 | 35 | −20 | 320 | 272,980 | 97,014 | 339,391 | 136,651 | 106,183 | 140,807 |
| 40 | 30 | 35 | −20 | 320 | 240,804 | 147,960 | 297,407 | 138,987 | 88,768 | 96,628 |
| 41 | 1 | 300 | −20 | 320 | 209,496 | 29,537 | 199,364 | 58,513 | 43,207 | 51,021 |
| 42 | 1 | 300 | −20 | 320 | 438,840 | 38,076 | 294,124 | 95,677 | 72,674 | 80,455 |
| 43 | 8 | 300 | −20 | 320 | 522,384 | 315,696 | 558,821 | 237,272 | 146,785 | 229,281 |
| 44 | 8 | 300 | −20 | 320 | 307,007 | 206,712 | 347,495 | 151,281 | 89,498 | 138,143 |
| 45 | 15 | 300 | −20 | 320 | 211,628 | 255,560 | 392,838 | 198,331 | 117,156 | 181,967 |
| 46 | 15 | 300 | −20 | 320 | 427,898 | 272,181 | 484,278 | 214,030 | 131,009 | 204,106 |
| 47 | 30 | 300 | −20 | 320 | 585,382 | 411,957 | 646,920 | 335,835 | 281,271 | 357,102 |
| 48 | 30 | 300 | −20 | 320 | 453,946 | 358,835 | 570,533 | 253,783 | 244,263 | 306,228 |
| 49 | 1 | 35 | 20 | 320 | 59,301 | 40,814 | 152,110 | 47,271 | 22,924 | 22,668 |
| 50 | 1 | 35 | 20 | 320 | 128,583 | 37,143 | 143,682 | 27,395 | 11,213 | 9617 |
| 51 | 8 | 35 | 20 | 320 | 134,804 | 13,087 | 97,797 | 25,040 | 28,257 | 74,345 |
| 52 | 8 | 35 | 20 | 320 | 72,361 | 39,568 | 125,790 | 49,748 | 26,338 | 42,522 |
| 53 | 15 | 35 | 20 | 320 | 202,770 | 26,348 | 132,699 | 59,221 | 38,731 | 59,152 |
| 54 | 15 | 35 | 20 | 320 | 268,232 | 27,753 | 171,495 | 68,128 | 67,211 | 118,823 |
| 55 | 30 | 35 | 20 | 320 | 36,275 | 5026 | 47,982 | 28,455 | 29,874 | 40,194 |
| 56 | 30 | 35 | 20 | 320 | 230,066 | 177,791 | 416,276 | 206,119 | 134,749 | 153,933 |
| 57 | 1 | 300 | 20 | 320 | 203,962 | 22,891 | 182,810 | 61,889 | 64,813 | 89,814 |
| 58 | 1 | 300 | 20 | 320 | 318,045 | 37,369 | 272,839 | 86,310 | 80,373 | 112,924 |
| 59 | 8 | 300 | 20 | 320 | 186,840 | 223,351 | 377,598 | 180,314 | 112,725 | 174,757 |
| 60 | 8 | 300 | 20 | 320 | 362,158 | 230,536 | 406,363 | 184,179 | 117,303 | 174,143 |
| 61 | 15 | 300 | 20 | 320 | 356,928 | 249,009 | 434,678 | 197,170 | 121,469 | 189,884 |
| 62 | 15 | 300 | 20 | 320 | 503,220 | 301,370 | 526,770 | 235,723 | 156,489 | 48,466 |
| 63 | 30 | 300 | 20 | 320 | 62,800 | 13,215 | 17,413 | 5716 | 4469 | 4931 |
| 64 | 30 | 300 | 20 | 320 | 377,885 | 239,466 | 389,722 | 193,828 | 163,154 | 200,208 |

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
