# Peer review of "Development, Optimization and Validation of a Sustainable and Quantifiable Methodology for the Determination of 2,4,6-Trichloroanisole, 2,3,4,6-Tetrachloroanisole, 2,4,6-Tribromoanisole, Pentachloroanisole, 2-Methylisoborneole and Geosmin in Air"

_processes, doi:10.3390/pr9091571_

Round 1

Reviewer 1 Report

The work presented for review is valuable and pleasant to read. The authors described the research problem in the introduction, methodology and research results in great detail. However, the authors do not discuss the obtained results with the data obtained by other authors. I advise you to pay attention to letter omissions, e.g. unit exponents should be given in superscript everywhere. Line 434: text: "Please add:" is unnecessary.

Author Response

Dear,

thank you for your comments. 

Response to Reviewer 1 Comments

1.1          the authors do not discuss the obtained results with the data obtained by other authors..

the diffusive sampling rate can be dependent on certain factors such as the compound or time of exposure for this reason its very difficult to compare with other studies. We found some studies such as Jia and Fu 2017 that used similar compunds and they obtained resulltas in the same order of magnitude. So, we only found one study that took into account one of the selected compounds (TCA) and obtained slightly lower results. we do not know why. More analysis is needed.

1.2          I advise you to pay attention to letter omissions, e.g. unit exponents should be given in superscript everywhere. Line 434: text: "Please add:" is unnecessary

Ok, the changes were done.

Reviewer 2 Report

The manuscript by Jové et al. deals with the development of a method for the quantification of four haloanisoles (2,4,6-Trichloroanisole, 2,3,4,6-Tetrachloroanisole, 2,4,6-Tribromoanisole, Pentachloroanisole) as well as of 2-Methylisoborneole and geosmin in the air of wineries.

The manuscript is interesting and well presented, thus it deserves to be published.

However, for the sake of completeness, I suggest including also the GC chromatograms reporting the peaks of the investigated compounds indicating the relative retention times and retention indices.

Author Response

Dear,

thank you for your comments. 

Response to Reviewer 2 Comments

2.1          However, for the sake of completeness, I suggest including also the GC chromatograms reporting the peaks of the investigated compounds indicating the relative retention times and retention indices.

Ok, the change was done.  

Reviewer 3 Report

The manuscript is difficult to read, because the objective was not well express and describe. There are so many discussions in the experimental part. And all the experiments were not sufficiently detailed, and the manuscript no well-organized.

First: The title is too long

Second : The authors claim a sustainable method. But there is no explanation for this claim and no scientific justification.

Third : there is a mixture between the thermal desorption method and the GC / MS method, while they should be treated as a single method, one does not go without the other. The GC / MS is there to measure the samples obtained by thermal desorption method. and the quantities measured by the GC / MS are adapted to the thermal desorption method. So optimization and validation must be done on the overall approach.

Eg line 11 to 114

“An experimental design was used to optimize the TD-GCMS method. For this rea son, a procedure to obtain tubes with known concentrations was used. The objective of the optimization was to find the best experimental condition (instrumental values) that maximized the detection of the six compounds simultaneously.”

And the section “2.3. Optimization and validation of the TD-GCMS method”  should be more segmented: instrumental part, design of experiment part, validation part, etc. And for example does the validation part should be in this paragraph?

A general control of abreviation and unit have to be done in this manuscript.

Line 103

TLC is the abbreviation for thin layer Chromatography, so don’t used it

Line 105

50°C seg-1

I don’t know this unit in the international system of unit

Line 108

“The Tenax TA tubes, spiked with a 5 mg L-1 of each analyte, were analysed by GC-MS methodology. The peak area of each compound was the response variable.”

During the optimization, does the molecules are tested independently or in mixture.

Lines 116 to 120

“The sampling end of the sorbent tube was connected to the CSLR system and the carrier gas flow rate was set at 80 ml min−1. The quantity of 1 µL of a mixed solution, containing all target compounds at concentrations between 0.05 mg L−1 and 5 mg L−1 and prepared in methanol, was in- troduced through the injector septum using a standard GC syringe.”

The units were not correctly written.

Lines 134 to 136

“GC oven program starting at 55°C (3 min), increased by 15°C/min to 125°C, 1.5°C/min to 145°C, 10°C/min to 183°C, 1.5°C/min to 195°C and 15°C/min to 250°C (held for 3 min).”

Please homogenize the way of units are written. Eg 15°C min-1

Line 144

Explain what do you mean to “recommended manufacturer values”

Table 1

Explain why the Trap heating rate doesn’t have to be optimized in your case. Demonstrate that was selected by Camino is transposable to your study. And explain ow this parameter (40 ml min-1 in line 150) which is not mentioned in the table 1. Same comment for the helium flow. Please use in the text the same abbreviation as in table 1, it should help to identify the parameters.

The validation procedure is not sufficiently describing. Eg How each samples were prepared, independent preparation, etc. In addition, Provide the reference of your methodology.  

And it is unclear what method is validated. Here the work describes a new method of thermal desorption-gas chromatography–mass spectrometry. So we expect the validation of this process. But the description is not sufficiently developed to be sure.

Explain the “response ratio” and “analyte peak ratio” in the sentence “ the response ratio of the analyte peak ratio was plotted as a function of analyte concentration and used to generate linear regression.” What are measure.

Line 157

For each compound the validation criteria must be evaluated in the presence of the other compounds in order to simulate real sample because compound interaction may influence the analytical response.

In you study you establish the intermediate precision not the reproducibility.

Line 164

“five replicates” mean five different tubes?

In equation 3 suppress the brackets which are not useful.

Line 204 and 205

“but humidity and temperature inside the chamber were also checked during assays using a thermo-hygrometer with a probe (TESTO 635)” provide the observed values or the selected values.

Line 217 to 219

“All tubes were enclosed and stored at 4°C until they were analysed using the GC-MS pro-cedure previously reported.” An also before the desorption step. So mention here which desorption conditions were used at this stage.

In Figures A1, A3, A5, A7, A9, A10, A 11 and A12 in Appendix A) and table A7, and table 4 specify the response (+ unit) observed in the model.

Part “3.2. GC-MS method validation”

The name of this part is not correct since the desorption approach must be also validated since the quality of the quantitative result approach of this work is based on this trap step.

Line 323 and 155

“0.1 to 2 ng at six concentration levels” weights are not concentration

“A set of six concentrations over an analytical range (from 0.1 to 2 ng) were analysed in triplicate for each compound.” Same comment.

The trueness which should be accuracy is not provided whereas it is mentioned in experimental section.

Table 5

The results are expressed in “(ng tube-1)” for LOD and LOQ and by “(ng)” for precision. So it is difficult to understand what is measured.

The validation is unclear, the objective and the results.

Line 343

“The behaviour of all analytes was the same: increasing over exposure time until it decreased.”

The curves are really different between compounds. Please show in figure 1 all the compounds on the whole time scale, you will see. Does the amount for each compound used in the experiment, is well selected?

And explain the high standard deviation of Mad.

For each compounds the time range chosen must be provided.

Author Response

Dear,

thank you for your comments.

Response to Reviewer 3 Comments

3.1          First: The title is too long

Maybe be for the name all the compounds. We believe it is necessary because they are not part of the same chemical family

3.2          Second : The authors claim a sustainable method. But there is no explanation for this claim and no scientific justification.

The method described is sustainable because, unlike the previous one, it does not use solvents. it is commented on the lines from 51 to 58.

3.3          Third : there is a mixture between the thermal desorption method and the GC / MS method, while they should be treated as a single method, one does not go without the other. The GC / MS is there to measure the samples obtained by thermal desorption method. and the quantities measured by the GC / MS are adapted to the thermal desorption method. So optimization and validation must be done on the overall approach.

Eg line 11 to 114. “An experimental design was used to optimize the TD-GCMS method. For this rea son, a procedure to obtain tubes with known concentrations was used. The objective of the optimization was to find the best experimental condition (instrumental values) that maximized the detection of the six compounds simultaneously.”

I do not understand this comment. In summary, we approach it as a single method with two steps: desorption conditions and chromatography conditions.

3.4          And the section “2.3. Optimization and validation of the TD-GCMS method”  should be more segmented: instrumental part, design of experiment part, validation part, etc. And for example does the validation part should be in this paragraph?

In this case, we consider that the section is already written in a segmented way, we did not want to add more subsections. We consider that the validation must be in the same section as the method being validated.

3.5           A general control of abbreviation and unit have to be done in this manuscript.

Thanks, we have already reviewed it.

3.6          Line 103. TLC is the abbreviation for thin layer Chromatography, so don’t used it

Thanks, the change was done.

3.7          Line 105. 50°C seg-1. I don’t know this unit in the international system of unit

Desorption equipment uses this units so we consider that it should be maintained

3.8           Line 108. “The Tenax TA tubes, spiked with a 5 mg L-1 of each analyte, were analysed by GC-MS methodology. The peak area of each compound was the response variable.” During the optimization, does the molecules are tested independently or in mixture.

Both.

3.9          Lines 116 to 120. “The sampling end of the sorbent tube was connected to the CSLR system and the carrier gas flow rate was set at 80 ml min−1. The quantity of 1 µL of a mixed solution, containing all target compounds at concentrations between 0.05 mg L−1 and 5 mg L−1 and prepared in methanol, was in- troduced through the injector septum using a standard GC syringe.” The units were not correctly written.

Thanks, the change was done.

3.10       Lines 134 to 136

“GC oven program starting at 55°C (3 min), increased by 15°C/min to 125°C, 1.5°C/min to 145°C, 10°C/min to 183°C, 1.5°C/min to 195°C and 15°C/min to 250°C (held for 3 min).” Please homogenize the way of units are written. Eg 15°C min-1

Thanks, the change was done.

3.10       Line 144. Explain what do you mean to “recommended manufacturer values”

Thanks, the change was done.

3.11       Table 1. Explain why the Trap heating rate doesn’t have to be optimized in your case. Demonstrate that was selected by Camino is transposable to your study.

The trap heating rate must be the maximum because we analyze volatile compounds. Camino also analyzes TCA and comments on this. This is discussed in the conclusions.

3.12       And explain ow this parameter (40 ml min-1 in line 150) which is not mentioned in the table 1. Same comment for the helium flow. Please use in the text the same abbreviation as in table 1, it should help to identify the parameters.

We used 40 ml min to improve the shape of the peaks. It may vary slightly depending on the equipment. In general, for other tests this flow is also used on the advice of the manufacturer.

3.13       The validation procedure is not sufficiently describing. Eg How each samples were prepared, independent preparation, etc. In addition, Provide the reference of your methodology.  

The preparation and number of samples is described in the material and methods section. (Lines 118 and 175)

3.14       And it is unclear what method is validated. Here the work describes a new method of thermal desorption-gas chromatography–mass spectrometry. So we expect the validation of this process. But the description is not sufficiently developed to be sure.

The validation parameters that are analyzed and exposed are those recommended by the Guidelines for Primary Validation Parameters. I do not understand why they are not enough.

3.15       Explain the “response ratio” and “analyte peak ratio” in the sentence “ the response ratio of the analyte peak ratio was plotted as a function of analyte concentration and used to generate linear regression.” What are measure.

I do not understand the comment. The quantification is done by calibration line, which is a very common method where the concentration and the area of the compound are taken into account. I don't think it is necessary to specify more.

3.16       Line 157. For each compound the validation criteria must be evaluated in the presence of the other compounds in order to simulate real sample because compound interaction may influence the analytical response.

For this reason real samples are analyzed.

3.17       In your study you establish the intermediate precision not the reproducibility

I don't know what you mean by intermediate precision. The term of precision is defined by the ISO International Vocabulary of Basic and General Terms in Metrology47 as the closeness of agreement between quantity values obtained by replicate measurements of a quantity under specified conditions. The determination of this parameter is done by the evaluating repeatability and reproducibility in the validation of the method. In this study, precision has been analyzed.

3.18       Line 164. “five replicates” mean five different tubes?

In the case of thermal desorption, it is not possible to analyze the same tube more than once. Analytes are desorbed in each assay. For this reason, the replicas correspond to different tubes. 

3.19       In equation 3 suppress the brackets which are not useful.

we consider keeping the brackets because it represents the R value

3.20 Line 204 and 205. “but humidity and temperature inside the chamber were also checked during assays using a thermo-hygrometer with a probe (TESTO 635)” provide the observed values or the selected values.

The observed values. The test is carried out at controlled temperature as discussed above.

3.20       Line 217 to 219

“All tubes were enclosed and stored at 4°C until they were analysed using the GC-MS pro-cedure previously reported.” An also before the desorption step. So mention here which desorption conditions were used at this stage.

We have specified TD-GCMS to make it clearer which are the conditions that have been optimized in the study.

3.21       Part “3.2. GC-MS method validation”

The name of this part is not correct since the desorption approach must be also validated since the quality of the quantitative result approach of this work is based on this trap step.

Correct, we have changed the title

3.22       Line 323 and 155. “0.1 to 2 ng at six concentration levels” weights are not concentration.  “A set of six concentrations over an analytical range (from 0.1 to 2 ng) were analysed in triplicate for each compound.” Same comment.

In this case it refers to ng tube-1. the changes were done.

3.23       The trueness which should be accuracy is not provided whereas it is mentioned in experimental section.

True, we have removed trueness from line 166

3.24       Table 5. The results are expressed in “(ng tube-1)” for LOD and LOQ and by “(ng)” for precision. So it is difficult to understand what is measured.

in this case it refers to ng tube-1. the changes were done.

3.25       The validation is unclear, the objective and the results.

We do not think that

3.26       Line 343. “The behaviour of all analytes was the same: increasing over exposure time until it decreased.” The curves are really different between compounds. Please show in figure 1 all the compounds on the whole time scale, you will see. Does the amount for each compound used in the experiment, is well selected?

The analyzed compounds are different, it is normal that they have different responses. The behaviour is the same for all as it happens in other studies.

3.27       And explain the high standard deviation of Mad.

we do not consider that they are so high.

Author Response

Dear,

thank you for your comments.

Response to Reviewer 4 Comments.

4.1          The authors have developed a method to monitoring and quantify important haloanisole compounds associated with cork and wine taint from the air, using passive sampling.

The methods applied in the study have been appropriately optimized and validated and will be useful for the alcoholic beverage industry.

I cannot find any issues with the introduction, methods, study design, results or discussion presented.  The only area in which the manuscript could be improved would be to move the plots and tables from the Appendix into a supplementary information file. This would make the manuscript tidier. 

Thanks for your comments. We will ask if it is possible to move the plots and tables from the Appendix into a supplementary information file.

4.2          Ensure that decimal places are denoted by '.' rather than ',' as in Figure 1.

We will do it. Thanks.

Reviewer 5 Report

The authors have developed a method to monitoring and quantify important haloanisole compounds associated with cork and wine taint from the air, using passive sampling.

The methods applied in the study have been appropriately optimized and validated and will be useful for the alcoholic beverage industry.

I cannot find any issues with the introduction, methods, study design, results or discussion presented.  The only area in which the manuscript could be improved would be to move the plots and tables from the Appendix into a supplementary information file. This would make the manuscript tidier. 

Ensure that decimal places are denoted by '.' rather than ',' as in Figure 1.

Author Response

Dear,

thank you for your comments.

Response to Reviewer 5 Comments.

5.1          The authors have developed a method to monitoring and quantify important haloanisole compounds associated with cork and wine taint from the air, using passive sampling.

The methods applied in the study have been appropriately optimized and validated and will be useful for the alcoholic beverage industry.

I cannot find any issues with the introduction, methods, study design, results or discussion presented.  The only area in which the manuscript could be improved would be to move the plots and tables from the Appendix into a supplementary information file. This would make the manuscript tidier. 

Thanks for your comments. We will ask if it is possible to move the plots and tables from the Appendix into a supplementary information file.

5.2          Ensure that decimal places are denoted by '.' rather than ',' as in Figure 1.

We will do it. Thanks.

Round 2

Reviewer 3 Report

Line 26 The results showed that the developed methodology is a sustainable and useful tool for the determination of these compounds in air.

Specify the results which demonstrated the sustainability of the method. In addition provide information about the reusability of the desorption tubes. And how is recycling.

Line 105. 50°C seg-1. So explain the abbreviation of “seg” and the meaning of the unit. Apparently this unit expressed a rate, so we expect a unit express with time.

Line 109. If the molecules are spiked in mixture it should be mention in the experimental and the status single or mixture mentioned in the results.

Line 168 Explain the “response ratio” and “analyte peak ratio” in the sentence “ the response ratio of the analyte peak ratio was plotted as a function of analyte concentration and used to generate linear regression.” What are measure.

“peak ratio” is unclear: what is calculated a ratio of heights, area “. And there is two ratio : the “response ratio of the analyte peak ratio”.

As mention in “the ISO International Vocabulary of Basic and General Terms in Metrology” there are three levels of precision: “ Measurement precision is used to define measurement repeatability, intermediate measurement precision, and measurement reproducibility.”

1repeatability

2intermediate precision

3reproducibility

In this work precision study correspond to levels 1 and 2. Reproducibility is for interlaboratory comparison.

So that should be corrected in the manuscript.

The quality of figure 1 is not sufficient.

A validation should include the determination of accuracy which was not performed in this work. So the term validation must be suppressed everywhere in this manuscript and replace by method performance.

Author Response

Dear reviewer 3,

3.1    Line 26 The results showed that the developed methodology is a sustainable and useful tool for the determination of these compounds in air.

Specify the results which demonstrated the sustainability of the method. In addition provide information about the reusability of the desorption tubes. And how is recycling.

Here are two aspects to consider:

  1. it is stainless steel tubes that are reused until the material breaks (it has never happened to us yet)
  2. it is a solvent-free methodology because the extraction process is carried out in the tube by high temperature.

For these two reasons we have considered the sustainable methodology

3.2    Line 105. 50°C seg-1. So explain the abbreviation of “seg” and the meaning of the unit. Apparently this unit expressed a rate, so we expect a unit express with time.

Seg is second. The change from ºC seg-1 to ºC s-1 was done.

3.3    Line 109. If the molecules are spiked in mixture it should be mention in the experimental and the status single or mixture mentioned in the results.

Experimental design: mixed solution (line 111)

Validation : single (line 168)

These conditions are mentioned in the materials and methods section. we believe that it is enough.

3.4    Line 168 Explain the “response ratio” and “analyte peak ratio” in the sentence “ the response ratio of the analyte peak ratio was plotted as a function of analyte concentration and used to generate linear regression.” What are measure.

“peak ratio” is unclear: what is calculated a ratio of heights, area “. And there is two ratio : the “response ratio of the analyte peak ratio”.

The change from “The response ratio of the analyte peak ratio was plotted as a function of analyte concentration and used to generate linear regression” to “The response of the analyte (area) was plotted as a function of analyte concentration and used to generate linear regression”

3.5    As mention in “the ISO International Vocabulary of Basic and General Terms in Metrology” there are three levels of precision: “ Measurement precision is used to define measurement repeatability, intermediate measurement precision, and measurement reproducibility.”

1repeatability

2intermediate precision

3reproducibility

In this work precision study correspond to levels 1 and 2. Reproducibility is for interlaboratory comparison. So that should be corrected in the manuscript.

The reproducibility is used for interlaboratory comparison but also to validate your method by analyzing the standards on different days.

3.6    The quality of figure 1 is not sufficient.

ok

3.7    A validation should include the determination of accuracy which was not performed in this work. So the term validation must be suppressed everywhere in this manuscript and replace by method performance.

Accuracy is a parameter that corresponds to validation, but even if it has not been tested, it does not mean that the method is not validated. Accuracy of this method would be measured by obtaining the recovery percentage of the compounds injected into the tubes at known concentrations. So, we found it much better to analyze the compounds in real samples.

thanks for your help.

Round 3
